# Andreev Blockade in a Double Quantum Dot with a Superconducting Lead

David Pekker and Sergey M. Frolov

*Department of Physics and Astronomy, University of Pittsburgh, Pittsburgh, PA, 15260*

A normal metal source reservoir can load two electrons onto a double quantum dot in the spin-triplet configuration. We show that if the drain lead of the dot is a spin-singlet superconductor, these electrons cannot form a Cooper pair and are blockaded on the double dot. We call this phenomenon Andreev blockade because it arises due to suppressed Andreev reflections. We identify transport characteristics unique to Andreev blockade. Most significantly, it occurs for any occupation of the dot adjacent to the superconductor, in contrast with the well-studied Pauli blockade which requires odd occupations. Andreev blockade is lifted if quasiparticles are allowed to enter the superconducting lead, but it should be observable in the hard gap superconductor-semiconductor devices. Andreev blockade should be considered in the design of topological quantum circuits, hybrid quantum bits and quantum emulators.

Transport blockade phenomena are interruptions of transmission due to interactions of multiple particles. They are a testbed for new physics related to coherence or conservation of charges, spins, photons and phonons [1–6]. The most iconic is the Coulomb blockade [7–11] which occurs when the energy barrier due to charging prevents electrons from tunneling through e.g. a quantum dot. Double quantum dots are known to demonstrate Pauli blockade due to spin-triplet states. This has been thoroughly studied in a large number of platforms, and is commonly used as an initialization and readout mechanism for quantum dot spin-based qubits [12–14]. The realization of single and double quantum dots coupled to superconductors, with induced Andreev Bound States [15–17], brings forward the question of whether there can be blockade phenomena specific to Andreev transport?

In this manuscript, we propose a blockade that appears in transport through a double quantum dot with at least one superconducting lead [18]. The gap in the superconductor prevents single-particle transport through the double dot. However, transport can still take place via the process of Andreev reflection in which two electrons from the double dot enter the superconducting lead as a Cooper pair. We find that if the two electrons have been loaded in a triplet state, Andreev reflection is suppressed. Both Pauli blockade and Andreev blockade involve triplet states of two electrons. However, the origin of the former is the Pauli exclusion principle that prevents electrons from passing through a dot already occupied by an electron of the same spin. The origin of Andreev blockade is angular momentum conservation: a pair of electrons must be in a singlet state in order to tunnel into the superconductor as a Cooper pair, hence electrons in a triplet state have the wrong total spin. While Andreev blockade happens at the dot-lead interface, it still requires two dots to manifest because the system needs to be filled into a low-energy (subgap) triplet state, e.g. with one electron on each dot (Fig. 1).

The transport signatures of Andreev blockade are summarized in Fig. 2. It is instructive to compare Andreev blockade to the well-studied Pauli blockade for the case of a double dot with normal leads. First, we observe that due to Coulomb blockade, transport is only allowed through the double dot in the vicinity of charge degeneracy points, which at finite source-drain voltage bias transform into double-triangle structures in the space of the two gate voltage that change the occupations of the two dots, $V_{g1}$ vs. $V_{g2}$.

Pauli blockade leads to suppressed conductance at the $(1,1)\rightarrow(0,2)$ charge degeneracy point, where (n,m) denote double dot occupations (Fig. 2(a)). Andreev blockade appears at the two $(1,\text{odd})\rightarrow(0,\text{even})$ charge degeneracy points (Fig. 2(b)), i.e. twice as often as Pauli blockade. Andreev blockade is only sensitive to the parity of the charging state of $QD_1$ due to the particle-hole symmetry in the superconductor. As in the case of Pauli blockade, changing the source-drain bias direction changes which charge degeneracy points are blockaded (see supplemental materials). In the case of Andreev blockade, switching the bias direction results in the blockade to the two $(1,\text{odd})\rightarrow(2,\text{even})$ charge degeneracy points.

A closer comparison reveals that the conductance triangles in Andreev transport (Fig. 2(b)) are approximately twice as large as in normal transport (Fig. 2(a)). At the same time, there is only one conductance triangle

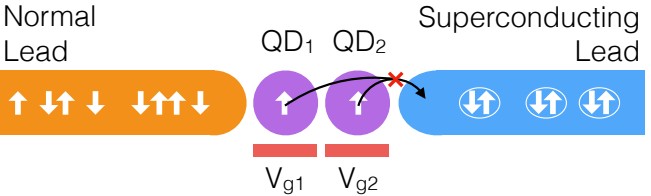

FIG. 1. Device schematic: Two quantum dots (labeled $QD_1$ and $QD_2$) are tunnel coupled to each other and to the two leads. The left, normal metal lead supports only single electron tunneling. The right, superconducting lead supports only Cooper pair tunneling. The chemical potentials on the quantum dots are tuned using two gates (labeled $V_{g1}$ and $V_{g2}$). Quantum dots are shown in the $T_+(1,1)$ configuration, but Andreev blockade also occurs for $T_0(1,1)$ and $T_-(1,1)$.

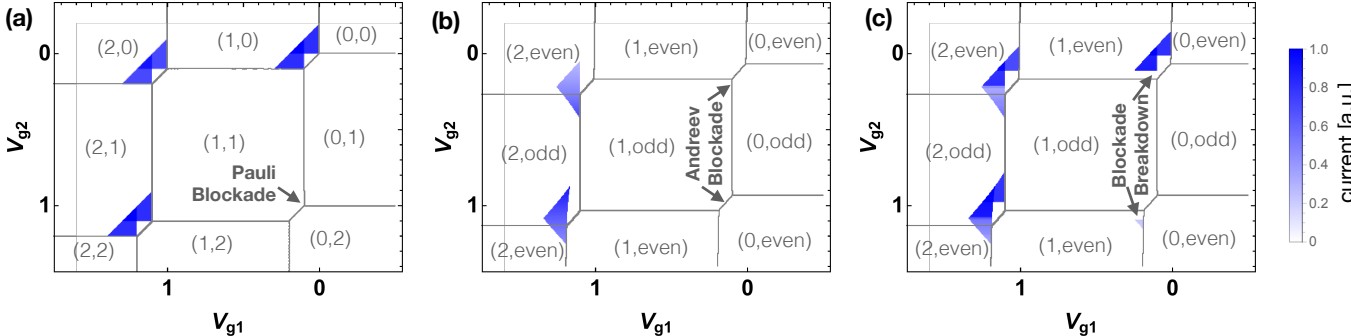

FIG. 2. Zero temperature charge stability diagrams comparing Pauli and Andreev blockades. Current through the double quantum dot at fixed source-drain bias voltage is plotted as a function of the gate voltages $V_{g1}$ and $V_{g2}$ on the two quantum dots. Gate voltages are in the units of dot charging energies $U_1 = U_2 = 1$. Labels in brackets indicate the lowest energy states of the double dot. (a) Pauli blockade: double quantum dot with two normal metal leads at low source-drain bias. (b) Andreev blockade: double quantum dot with one normal metal lead and one superconducting lead at low source-drain bias. (c) Breakdown of Andreev blockade: when the source drain bias exceeds the superconducting gap (which we reduce from infinite to $0.05U_1$ in this plot), the conductance triangles partially reappear. Other parameters used: inter-dot charging energy $U_{12} = 0.1$, Andreev coupling $\Delta_2 = 0.25$, source and drain bias $V_1 = -V_2 = 0.1$, further details in the supplemental materials.

at each charge degeneracy point in Andreev transport, but two in normal transport. Furthermore, Fig. 2(c) shows that the breakdown of Andreev blockade at large source-drain bias is different from Pauli blockade: in Andreev blockade the source-drain bias needs only to exceed the superconducting gap of the lead in order for breakdown to occur, while in Pauli blockade the bias must exceed the (0,2) singlet-triplet energy (not shown) [12].

In order to understand these manifestations of the Andreev blockade, let us first consider a single quantum dot coupled to a superconducting lead. In the absence of Andreev reflection, the quantum dot can be in one of four states: $|0\rangle$, $|\uparrow\rangle$, $|\downarrow\rangle$, or $|\uparrow\downarrow\rangle$ (corresponding to empty, spin-up electron, spin-down electron, and doubly occupied). Conventionally, the charging state of the dot is denoted as $\{0, 1, 2\}$ (where the number indicates the number of electrons on the dot). As Andreev reflection mixes the two even parity states $|0\rangle$ and $|\uparrow\downarrow\rangle$ we switch over to parity notation $\{\text{even}, \text{odd}\}$ to denote the state of a quantum dot coupled to a superconducting lead. Starting from an odd parity state we can reach an even parity state by either adding or removing an electron. Hence, the levels of the quantum dot coupled to a superconducting lead can be thought of as approximately particle-hole symmetric.

The mixing of the empty and doubly occupied states implies that the two charge degeneracy points of the quantum dot nearest to the superconducting lead are equivalent. Therefore, Andreev blockade can only be controlled by the occupancy of the quantum dot nearest to the normal lead, which is the reason why Andreev blockade occurs twice as often as Pauli blockade.

The approximate particle-hole symmetry implies that the conductance is approximately unchanged as the quantum dot nearest to the superconducting lead is tuned from slightly above the charge degeneracy point to slightly below it. That is, conductance can take place on both the particle-like and hole-like side of the charge degeneracy point. On the other hand, the version of the device with two normal metal leads can only support conductance on the particle-like side of the charge degeneracy point. The additional hole-like conductance regime that is present in Andreev transport results in the approximate doubling in the size of conductance triangles in Andreev as compared to normal transport.

The fact there is only one transport triangle at each charge degeneracy point in Andreev transport (Fig.2(b)), while there are two in normal transport (Fig.2(a)), is a consequence of the fact that in Andreev transport two electrons must be moved from the source to the drain lead per transport cycle, while only one electron is moved in normal transport. Normal charge transport cycle goes through three charging states and hence requires a triple charge degeneracy point (i.e. a point at which three charging states become degenerate). The Andreev transport cycle goes through four charging states and hence requires a quadruple charge degeneracy point. In normal transport, finite inter-dot coupling splits the quadruple charge degeneracy point into two triple charge degeneracy points and hence the number of conductance triangles doubles resulting in characteristic hexagonal patterns of charge transport in double quantum dot systems. On the other hand, the number of triangles in Andreev transport remains unchanged as all four charging states are required for transport.

*Andreev charge transport cycle* – We use the master equation formalism to describe electron transport (see supplemental materials for details of the method). Our strategy is to begin by considering the eigenstates of $QD_1$ and of $QD_2$ independently. We assume weak interdot

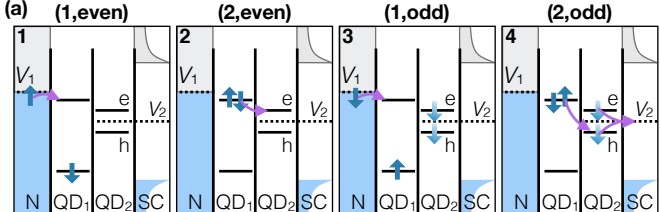 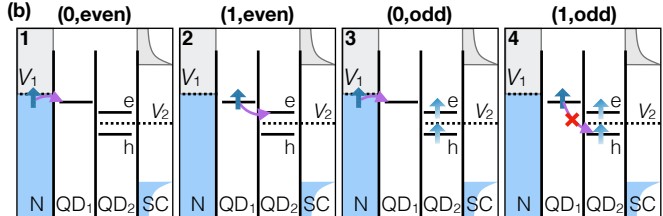

FIG. 3. (a) Andreev transport cycle. $QD_1$ is tuned to the 1-2 charge degeneracy point, while $QD_2$ is tuned to the vicinity of one of the charge degeneracy points, such that even state is the ground state. The transport cycle consists of four charge states depicted in the Figure. The incoherent electron tunneling processes (which we describe using the master equation formalism) are depicted by purple arrows. Odd parity states of $QD_2$ are depicted as superpositions of electron and hole states to denote the approximate particle-hole symmetry (i.e. the fact that odd $\rightarrow$ even transition can occur via either electron addition or electron subtraction). (b) Andreev blockade. Setup identical to panel (a) except the $QD_1$ is tuned to the 0-1 charge degeneracy point. Andreev blockade occurs in the last step: $QD_1$ and $QD_2$ both host a spin-up electron, consequently the spin-up electron on $QD_1$ cannot tunnel onto $QD_2$ to make a Cooper pair.

tunnel coupling so that the double dot states are well approximated by products of the eigenstates of the two dots. The charge transport cycle involves sequential incoherent transitions between $QD_1$ and $QD_2$ eigenstates.

To construct the eigenstates of a quantum dot Andreev-coupled to a superconducting lead ($QD_2$) we use the Hamiltonian

$$
H_{\text{QD-Andreev}} = \begin{pmatrix} 0 & 0 & 0 & \Delta \\ 0 & \epsilon_\uparrow & 0 & 0 \\ 0 & 0 & \epsilon_\downarrow & 0 \\ \Delta & 0 & 0 & \epsilon_\uparrow + \epsilon_\downarrow + U - 2eV \end{pmatrix}, \quad (1)
$$

where $\epsilon_\uparrow$ and $\epsilon_\downarrow$ are the single-electron energies, $U$ is the quantum dot charging energy, $\Delta$ is the Andreev reflection amplitude, and $2eV$ is the energy of one Cooper pair in the superconducting lead biased to voltage $V$. The eigenstates that play a role in transport are the two odd parity eigenstates ($|\uparrow\rangle$ and $|\downarrow\rangle$) and the lower energy even parity eigenstate ($|e\rangle$) that is the superposition of the states $|0\rangle$ and $|\uparrow\downarrow\rangle$.

Each Andreev charge transport cycle adds a Cooper pair to the superconducting lead. We first consider a cycle without Andreev blockade (Fig. 3(a)). $QD_1$ is tuned to the charge 1-2 degeneracy point, with the 1 state being slightly lower in energy, while $QD_2$ is tuned to the even-odd degeneracy point with the even state being slightly lower in energy. The transport cycle consists of four steps: (1) an electron from the normal lead moves onto $QD_1$; (2) an up-spin electron moves from $QD_1$ to $QD_2$ resulting in $QD_2$ being excited into the odd state; (3) another electron from the normal lead moves onto $QD_1$; (4) a down-spin electron from $QD_1$ moves to $QD_2$ brining $QD_2$ back to the even ground state. Crucially, the two electrons that entered the double dot system from the left lead in steps (1) and (3) are absorbed into the right lead as a Cooper pair in step (4).

Gating $QD_1$ to the 0-1 charge degeneracy point results in Andreev blockade, which is illustrated in Fig. 3(b). The transport cycle proceeds through the same steps, but

becomes stuck at step (4) as an inter-dot triplet, that is incompatible with Andreev reflection, is formed on step (3).

*Breakdown of Andreev blockade* – Andreev blockade breaks down when single electrons are allowed to tunnel into the superconducting lead as quasiparticles. Quasiparticle excitations become important in two experimentally relevant cases. First, at sufficiently high source-drain bias the quasiparticle sates above the superconducting gap become accessible (Fig.2(c)). Second, superconductors can have low-energy quasiparticles due to nodes in the order parameter, vortices, or disorder. Andreev blockade is also lifted by any spin mixing mechanism, such as due to hyperfine, spin-orbit or electron-phonon interactions [19–21]. Since spin mixing in double dots has been studied previously and is not specific to superconductors, here we focus on quasiparticle transport.

We model a single superconducting lead with a quasiparticle density (a gapless superconductor) using a two-lead model, following Ref. [22]. The first virtual lead describes Cooper pair tunneling, and we model its effect on the adjacent quantum dot using Eq. (1). The second virtual lead describes single particle transport into the superconductor, and is modeled as a normal metal with a variable density of states. The tunneling of single electrons between the quantum dot and the second virtual lead is assumed to be an incoherent process, which we model at the master equation level.

Let us now consider transport in the (normal lead)-(double quantum dot)-(gapless superconducting lead) setup. Naively, we expect that transport can occur either through Andreev reflection or through normal single-particle transport and hence we can find the total current by adding up the two contributions (i.e. superimposing Figs. 2(a) and (b)). Transport calculations (Fig. 4(a)) are largely consistent with this notion, for example the upper right charge degeneracy point is no longer blockaded, and the triangles are doubled at each degeneracy point.

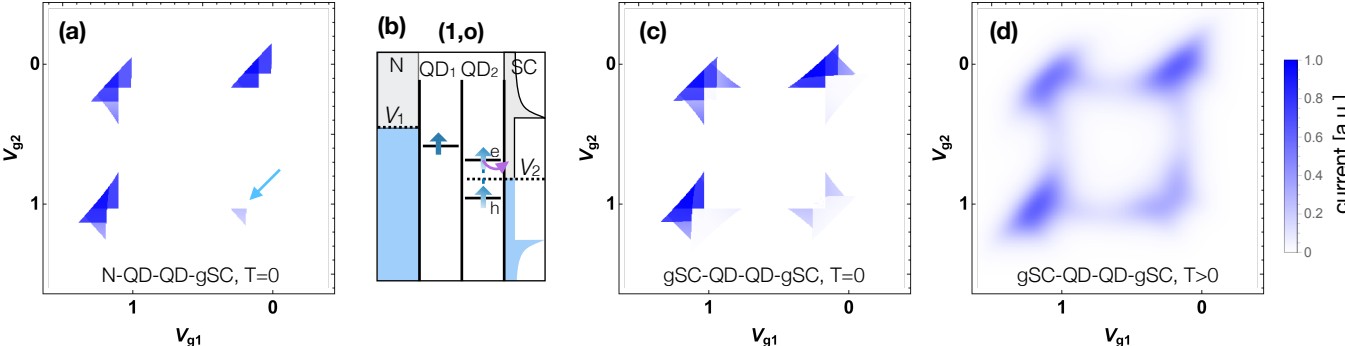

FIG. 4. Interplay of Andreev and spin blockade. (a) Charge stability diagram for a N-QD-QD-(gapless SC) system showing a transport pattern consisting of conventional transport features, Andreev transport features, and a new feature indicated by the blue arrow. The new feature is a consequence of tunneling process depicted in (b). (b) Tunneling process enabled by quasiparticles in the superconducting leads that results in the clearing of the inter-dot triplet. (c) Charge stability diagram for a (gapless SC)-QD-QD-(gapless SC) system. (d) Same as (c) but at $T = 0.05U_1$.

There is also a new feature: transport is allowed in the bottom right corner that was blockaded both in Andreev and Pauli cases. The single electron tunneling processes opens a pathway for clearing the inter-dot triplet state as illustrated in Fig. 4(b). Due to the finite density of states near the chemical potential spin-up Andreev bound state can leak out into the lead.

The zero- and finite-temperature charge stability diagrams for the case in which both leads are gapless superconductors are depicted in Figs. 4(c) and (d). The interplay of normal and Andreev transport results in an intermediate charge stability diagram with current at all four charge degeneracy points. At zero temperature, the charge stability diagrams with two gapless superconducting leads Fig. 4(c) and two normal metal leads Fig. 2(a) are clearly distinguishable. At finite temperature the distinction becomes blurred and in general no strong blockade of either kind is observed. The bottom right degeneracy point still shows lower current than the other three. Fig.4(d) closely matches recent data on double quantum dots with two gapless superconducting leads [22], where this regime has been interpreted as Pauli blockade based on blockade lifting due to spin-orbit interaction observed at finite field.

*Conclusions and outlook* – We have proposed a transport phenomenon that occurs in double quantum dots with a superconducting lead. The origin of the proposed Andreev blockade is that a low-lying triplet state suppresses Andreev reflection because electrons from the double dot cannot tunnel into the superconductor as a Cooper pair. The experimental consequence of Andreev blockade is the interruption of transport at two of the four charge degeneracy points (as compared to one of the four for Pauli blockade).

Quantum dots coupled to superconductors are at the crossroads of several promising research directions such as topological quantum computing, hybrid superconduct-

ing quantum bits [23, 24] and quantum simulation. Given the high interest in this system, Andreev blockade is likely to be observed. The key experimental technology required is to combine the quantum dot setup with a hard gap superconductor. These systems are already developed in efforts to realize Majorana zero modes in hybrid superconductor-semiconductor devices [25–27].

Proposed topological quantum computing architectures feature single and double quantum dots for Majorana state readout [28–30]. Superconducting double dots are investigated in the context of crossed Andreev reflection which is a key ingredient in recent parafermion proposals [31, 32]. Finally, chains of superconducting quantum dots have been proposed as emulators of the one-dimensional Kitaev model [33–35]. Andreev blockade phenomena may manifest in all of the above situations and can be leveraged to enhance advanced quantum device functionality.

*Acknowledgements* – We thank A. Tacla for insightful discussions. DP and SMF acknowledge support from the Charles E. Kaufman foundation, NSF PIRE-1743717. SMF acknowledges NSF DMR-1252962, NSF DMR-1743972, ONR and ARO.

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
