# Peer review of "Theory of Andreev Blockade in a Double Quantum Dot with a Superconducting Lead"

_SciPost Physics_

## Round 2 · Referee Report · Anonymous · 2021-9-20

Strengths

1- It describes a simple transport blockade mechanism that is relevant and readily detectable in double dot/superconductor junctions
2- The presentation is very to-the-point, and immediately addresses the obvious question of the difference to the better known spin-blockade mechanism
3- It references experiments demonstrating the effect
4- It contains microscopic rate-equation computations and an analysis of some of the blockade breakdown mechanisms

Weaknesses

1- Although not in this specific form, there are prior reports of analogous mechanisms, such as that in Ref [26], so the novelty is not so high
2- It describes a somewhat trivial effect (few would call it "groundbreaking"), although it remains an interesting twist on the spin/charge blockade story

Report

I found the description of the Andreev blockade mechanism very clear and instructive, especially in regards to the comparison to spin blockade. It is not particularly striking or physically revealing, but it does complement our vocabulary of transport phenomena with a new useful term. Beyond discussions of their significance, all the claims seem valid on a first read. The results might not be 100% original, since the mechanism is so obvious that anybody dealing with double dot in series with a superconductor is bound to have encountered this. The authors themselves mention Ref [26] which is rather close. This paper was posted in the arXiv before Ref [26], but the latter has already been published. Nothing to add on the subject of presentation, it is perfectly adequate and focused.

Requested changes

I cannot suggest any improvements to the manuscript, as formally it is complete and well crafted. The only problem is with originality and significance, which cannot be addressed by incremental improvements.

---

## Round 2 · Referee Report · Anonymous · 2021-9-24

Strengths

1. clear presentation of the transport phenomenon in question
2. immediate experimental relevance
3. several interesting detailed predictions

Weaknesses

1. While it is a paper on theory, there is virtually no theoretical analysis (see the report)
2. Minor shortcomings in presentation/reasoning.
3. Similar transport phenomena have been studied\discussed in the literature.

Report

Let me immediately report that the paper does not satisfy the acceptance criteria of SciPost. This is not because the paper is bad: it is rather about the criteria. To require from a good paper to report on "groundbreaking results" is either hypocrisy or madness. If it were realistic, in short time we would not have any ground to break. Anyway, I was asked to check the criteria, and this paper does not fit those. Perhaps SciPostPhysCore is a better destination.

This paper clarifies a transport phenomenon and gives name to it. Owing to the conceptual simplicity of the effect, similar transport features have been discussed in the literature the authors cite. Yet nobody has presented it with such concentation and detail. In contrast to the opinion of another referee, I do not think the Ref 26 did so.

Although the paper is on theory, the quantitative theoretical analysis is virtually absent: there are only 2 formulas in the paper representing the Hamiltonian, and many qualitative statements. All of them are relevant, and perhaps the simplicity of the effect makes the theoretical analysis rather redundant.

Requested changes

I've found 3 statements of secondary importance that do not sound good to me, perhaps because of the formulation. I would suggest the authors either correct or explain in more detail.
1. "The origin of Andreev blockade is angular momentum conservation:" There is no angular momentum conservation in the setup since the potential experienced by electrons is rather far from being spherically symmetric. The authors might mean spin conservation.
2. "While Andreev blockade happens at the dot-lead interface" I do not understand the use of "interface" here. Is there a statement about a physical location of the phenomenon? Not sure such statements are relevant for quantum particles.
3. "it still requires two dots to manifest because the system needs to be filled into a low-energy (subgap) triplet state, e.g. with one electron on each dot"
For me, two dots is thus a convenient model. Two dots in series can be always regarded as a single dot with, say, two levels. My guess is that the authors require a triplet ground state of either single or double dot, rather than two dots.

  • validity: high
  • significance: good
  • originality: high
  • clarity: top
  • formatting: excellent
  • grammar: perfect

Author:  Sergey Frolov  on 2021-09-24  [id 1782]

(in reply to Report 2 on 2021-09-24)

We thank both referees for inputs and considerations. We are satisfied that both referees find the conclusions to be valid and point out no errors. We find this the most important aspect of the peer-review-process.

The referees differ on their evaluation of the Significance and Originality. Referee 1 marks both as ‘OK’, and Referee 2 marks them as ‘Good’ and ‘High’.

We point out that this paper is an ‘Experimental Proposal’ theory paper. The value and significance of these type of papers is that they aim to put forward a theory that can actually be tested in experiment. It is highly valuable to think not only of new models or concepts, but also about what can be realized in the lab. The reality is that a majority of proposal papers are not followed up by experiments. In the case of this paper, it has been first made public on arxiv in 2018. This year, in 2021, our group has conducted experiments that test the theory (P. Zhang, H. Wu, J. Chen, S. A. Khan, P. Krogstrup, D. Pekker, and S. M. Frolov, arXiv:2102.03283 (2021)). The experimental paper is still under peer review, but even the fact that it was possible to do the experiment highlights the relevance and impact of the theory provided here. We did not see an obvious reference to the fact that the experimental paper exists in either referee report so we wanted to make sure that the referees are aware of this. The focus on doable experiments also dictates why the paper is heavy on predictions of data and not on the formulas. The paper in fact analyzed a dynamical phenomenon of the blockade which is best handled by numerical simulations.

We feel obliged to separately comment on the statement by Referee 2 about the acceptance criteria at SciPost. While we understand the comment as not aimed at our paper, but as a comment aimed at the journal, we feel that it may influence the decisions by the editor.

We point out that the requirement is to satisfy 1 out of 4 acceptance criteria. And we argue that criterion #3 applies here:

  1. Open a new pathway in an existing or a new research direction, with clear potential for multipronged follow-up work;

As we point out above, this theory published in version 1 on arxiv in 2018, has led to a follow-up experiment. We are also planning to do more experiments in a new materials system. And there have been other triplet blockade experiments done on superconducting devices in the meantime. So the follow-up work is multi-pronged. There is no need to guess whether this will happen, it has already happened. A new pathway has been opened.

Referee 2 highlights 3 sentences are requests clarification. We will re-upload a clarified version to arxiv, but since we are only given 1 week to respond we comment here so that the editorial college could hear our point of view before their vote.

  1. "The origin of Andreev blockade is angular momentum conservation:" There is no angular momentum conservation in the setup since the potential experienced by electrons is rather far from being spherically symmetric. The authors might mean spin conservation.

Response: in the presence of spin-orbit coupling spin is not a good quantum number. Angular momentum is more general and includes spin and orbital momentum. We could change it to ‘spin’ but in the past reviewers in such cases were concerned that this was not accurate. Though we agree that ‘spin’ is more pedagogical and works for most readers.

  1. "While Andreev blockade happens at the dot-lead interface" I do not understand the use of "interface" here. Is there a statement about a physical location of the phenomenon? Not sure such statements are relevant for quantum particles.

Response: What actually happens upon Andreev Blockade is Andreev reflection is suppressed. Andreev reflection is conventionally discussed as an interface property, between the superconductor and the normal conductor. Yes quantum particles do occupy the entire Universe, each of them does. We could clarify this, and improve the clarity of the sentence by pointing out that it is Andreev reflection at the N-S interface that is getting suppressed.

3. "it still requires two dots to manifest because the system needs to be filled into a low-energy (subgap) triplet state, e.g. with one electron on each dot" For me, two dots is thus a convenient model. Two dots in series can be always regarded as a single dot with, say, two levels. My guess is that the authors require a triplet ground state of either single or double dot, rather than two dots.

In a single quantum dot the triplet state will occupy a higher orbital. And the singlet-triplet energy often exceeds the superconducting gap. Ground state triplet can be obtained at finite magnetic field or in the presence of the magnets. This situation has been heavily studied for several decades. It is well known that in this case induced superconductivity is suppressed. Andreev blockade is a dynamic phenomenon: singlet and triplet are degenerate and both can form. But after a triplet is formed, the particles are stuck. This requires a low energy triplet state and a double quantum dot, so that each of the spins occupies its own orbital and the orbital energies are degenerate.

This report is prepared by S. Frolov

---

## Editorial Decision

publication_decision_taken:_accept